# Field Experience of Antibody Testing against *Mycoplasma bovis* in Adult Cows in Commercial Danish Dairy Cattle Herds

**DOI:** 10.3390/pathogens9080637

**Published:** 2020-08-06

**Authors:** Mette Bisgaard Petersen, Lars Pedersen, Lone Møller Pedersen, Liza Rosenbaum Nielsen

**Affiliations:** 1Department of Veterinary Clinical Sciences, University of Copenhagen, Agrovej 5A, 2630 Taastrup, Denmark; 2SEGES, 8200 Aarhus, Denmark; larp@seges.dk; 3Eurofins Milk Testing Denmark, 6600 Vejen, Denmark; lonemollerpedersen@eurofins.dk; 4Department of Animal and Veterinary Sciences, University of Copenhagen, Grønnegårdsvej 8, 1870 Frederiksberg, Denmark; liza@sund.ku.dk

**Keywords:** *Mycoplasma bovis*, diagnosis, control, immune response, ELISA

## Abstract

*Mycoplasma bovis* in cattle is difficult to diagnose. Recently, the ID screen^®^ mycoplasma bovis indirect ELISA (ID screen) was commercially released by IDVet. The objectives of this study were to: (1) gain and share experience of using the ID screen in adult dairy cows under field conditions; (2) determine the correlation between antibody levels in milk and serum and (3) compare the ID screen results with those of the Bio K 302 (BioX 302) ELISA from BioX Diagnostics. Paired serum and milk samples were collected from 270 cows from 12 Danish dairy herds with three categories of *M. bovis* disease history. The ID screen tested nearly all cows positive in all, but the three non-infected herds, while the BioX 302 tested very few cows positive. The ID screen is therefore a much more sensitive test than the BioX 302. However, cows in five exposed herds without signs of ongoing infection and two herds with no history of *M. bovis* infection also tested ID screen positive. Therefore, the performance and interpretation of the test must be investigated under field conditions in best practice test evaluation setups. A concordance correlation coefficient of 0.66 (95% CI: 0.59–0.72) between the ID screen serum and milk results indicates that milk samples can replace serum samples for the ID screen diagnosis of *M. bovis* in adult cows.

## 1. Introduction

*Mycoplasma bovis* (*M. bovis*) is an emerging bacterium associated with disease in cattle of all ages in many countries around the world [1]. In dairy cows, the usual presentation is mastitis, pneumonia and/or arthritis, while calves typically suffer from pneumonia, otitis media and/or arthritis [2,3]. Diagnosing *M. bovis* is challenging at both animal and herd level. *M. bovis-*associated disease can be diagnosed by using bacterial culture or PCR on body fluids or organ specimens and antibody measurements in serum or milk [3]. However, the fact that *M. bovis* bacteria lead to so many different disease manifestations and varying test responses in different age groups, and the fact that there is not one single diagnostic material that can test for and differentiate between all these disease manifestations makes it difficult to diagnose *M. bovis-*associated disease [4].

Antibody tests are inexpensive and for some purposes, it is an advantage that they can also detect previous (recent) infection. The first and previously only commercially available test for antibodies directed against *M. bovis* was produced by BioX Diagnostics in Belgium. The Bio K 302 ELISA kit (BioX 302) has been reported to have low sensitivity ranging from 0.37–0.50 and specificity ranging from 0.90–0.96 in experimental studies [5,6,7] and very short-lasting antibody detection in individual cows [8] and calves [9]. Petersen et al. [8] found that the mean antibody level in cows with clinical indication of *M. bovis* was only above the recommended cutoff (37 ODC%) for approximately 60 days after the disease outbreak, which implies that frequent testing would be necessary to detect disease among cows if the BioX 302 were to be used to assess the *M. bovis* status of dairy cows or herds. One explanation could be the large antigenic variation of *M. bovis* and the alterations of membrane surface lipoproteins over time [10]. Studies have compared the BioX 302 to an in-house *M. bovis* ELISA based on a different antigen. The agreement between the results from the two tests was low, and the antibodies detected by the in-house ELISA persisted in serum from cows 1.5 years after the disease outbreak regardless of the current *M. bovis* clinical status [9,11]. Results from one ELISA test can therefore not be extrapolated to other *M. bovis*-detecting ELISAs. The differences in test performance may be influenced by the different antigens used in each ELISA, how immunogenic they are, how long the immune system reacts to the particular protein and how similar the gene of that particular protein is in different *M. bovis* strains.

The ID screen^®^ mycoplasma bovis indirect (ID screen) from IDvet (Grabels, France) is a reasonably new commercially available antibody test. According to the manufacturer, the diagnostic sensitivity and specificity are 95.7% and 100%, respectively [12]. The test has only been evaluated in calves [12], but the age of the animals is very likely to influence the test performance when used under field conditions [13,14]. Therefore, current knowledge about the ID screen test performance may not be valid for adult cows and may vary depending on whether the test material is serum or milk.

Applying the BioX 302 to herd-level testing using bulk tank milk has been evaluated and found useful in estimating the prevalence at a national level when the cutoff was raised to 55 ODC% [15]. However, the challenge is that bulk tank milk primarily reflects *M. bovis* udder infections in the herd [8]. In fact, one study found that the hospital herd was the most indicative group to use for the detection of herd-level *M. bovis* infection based on bulk tank milk tested using the BioX 302 [16]. Danish cattle farmers have experienced many *M. bovis* disease outbreaks characterized by arthritis rather than mastitis as the primary clinical sign [8], and bulk tank milk samples would most likely fail to detect these outbreaks. Use of the BioX 302 on milk samples is non-optimal due to the need for frequent testing to ensure infected herds are detected (e.g., for classification of herds in relation to trade, shows, etc.), and because not all disease manifestations can be detected when measuring antibodies in milk [4,16]. As the ID screen is more sensitive than BioX 302, antibody measurements in milk may be more reliable, potentially making it feasible to use antibody testing of individual and bulk tank milk samples for surveillance or outbreak diagnostics, providing the specificity is sufficiently high. The potential use of ELISA on milk samples to classify or monitor dairy herds for *M. bovis* infection will be of interest in a setting like the Danish dairy industry, since the sampling can be automated via the mandatory milk quality control scheme and bulk tank milk surveillance for other cattle diseases.

The objectives of this study were therefore to: (1) gain and share experience of using the new commercial ELISA ID screen for the detection of antibodies against *M. bovis* in adult dairy cows under field conditions; (2) determine the correlation between the measured antibody levels in milk and serum and (3) compare the ID screen results with the results of the frequently used and commercially available BioX 302 ELISA.

## 2. Results

Paired serum and milk samples were collected from a total of 270 cows from 12 Danish dairy herds. All nasal swabs and milk samples from lactating cows in the Robust Calves herds (RC-herds) tested negative for the presence of *M. bovis* by PCR. See Table 1 and the Section 4 for a description of the different herds included in the study.

The serum and milk ID screen sample-to-positive percentage (S/P%) was plotted with jittered dots for each of the 12 herds (Figure 1 and Figure 2). All cows had a S/P% below the recommended cutoff in both serum and milk in three herds (herds 1–3). In all other herds, all or nearly all cows had a S/P% above the recommended cutoff.

The serum and milk BioX 302 sample-coefficient (ODC%) was plotted with jittered dots for each of the 12 herds (Figure 3 and Figure 4). All, but 16 cows from seven different herds had serum ODC% values below the recommended cutoff in serum, while all, but 17 cows from nine different herds had milk ODC% values below the recommended cutoff.

### Correlation between Serum and Milk S/P%

The concordance correlation coefficient between serum and milk S/P% across the full dataset of paired serum and milk samples was 0.66 (95% CI: 0.59–0.72). The correlation between serum and milk samples within each herd is shown in Figure 5. Correlations are not shown for the BioX 302 due to the low number of positive samples.

## 3. Discussion

In the present study, we tested milk and serum samples from cows in Danish dairy herds for which we had prior knowledge of the *M. bovis* infection status, in order to gain and share experience of using the ID screen for detecting antibodies against *M. bovis* in adult dairy cows under field conditions and to compare the results with the results of BioX 302. We also determined the correlation between the measured antibody levels in milk and serum for the ID screen across the full dataset of 12 dairy herds and inspected visualizations of the different correlation patterns for the individual herds.

### 3.1. Field Performance of the ID Screen and BioX 302

Overall, many cows and most of the herds tested positive in both serum and milk when the ID screen was used (Figure 1 and Figure 2). In only three of the 12 herds (Herds 1–3), all samples were negative in both serum and milk. These herds were all previously classified as not infected with *M. bovis,* indicating good concordance between the classification and the test results for these three herds. However, herds 4 and 5 were also classified as not infected, but most both serum and milk samples were positive. This is interesting and there could be several reasons for this, as discussed below.

With regard to the five herds classified as infected with *M. bovis* within the last 5 years (Herds 6–10), nearly all cows tested positive using the ID screen, despite the fact that none of the farmers thought that they had ongoing disease problems related to *M. bovis* at sampling. Four of the herds had an *M. bovis* disease outbreak 4–5 years prior to sampling. Most sampled cows in the three Outbreak-herds had not been born at the time of the disease outbreak. The persistence of antibodies in these herds therefore suggests that the cows were still exposed to *M. bovis*, despite not showing clinical signs around the time of sampling in this study. The two RC-herds classified as infected within the last 5 years both had calves that tested positive for antibodies in both the ID screen and BioX 302, as well as positive PCR samples. In these herds, it is more apparent that the animals were probably exposed to *M. bovis* around the time of sampling. If milk samples and nasal swabs for PCR-testing had been collected from the Outbreak-herds, it cannot be ruled out that some of them would have been positive as well, indicating a recent exposure to *M. bovis* despite there being no sign of disease.

The two herds classified as infected at the time of sampling both had all, but one sample above the cutoff and some of the highest S/P% seen in this study. This makes good sense in terms of the classification, and there was good concordance between the classification and test results in these herds. There was ongoing *M. bovis* infection at least among the calves, where the positive PCR samples were collected. However, none of the PCR tests from nasal swabs or milk samples collected from cows were positive. These were samples from healthy cows, and it cannot be ruled out that if the same samples had been collected from diseased cows, an indication of *M. bovis* infection may have been observed among the cows [4,16].

None of the study herds had ongoing clinical signs of an *M. bovis* disease outbreak. However, the herds still tested positive using the ID screen. All sampled cows were considered to be healthy by the farmer, were housed in the main milking herd and delivered milk to the bulk tank on the day of sampling. Based on this, it is not possible to use the ID screen to differentiate between healthy and diseased cows, but it is likely that the ID screen tests for exposure to *M. bovis*. Cattle can be subclinically infected with *M. bovis* [3], and if the ID screen tests positive in subclinically infected animals, it is potentially a very useful test to use in relation to prevent the spread of infection. However, further studies are needed as this study did not determine the *M. bovis* status of the individual cow, but the herd as a whole.

The ID screen is a sensitive test, as all or nearly all cows in each herd tested either positive or negative. This makes the test good for herd-level control and surveillance purposes, as a small sample of cows would give a good indication of the exposure status of the age group as a whole. However, there may be issues with the diagnostic (field-use) specificity of ID screen, as many of the cows in two out of the five herds classified as not infected within the previous 5 years prior to sampling did test positive. A possible explanation could be that the tested cows in the herds classified as non-infected had been subclinically infected [3] and had therefore never shown any clinical signs. The historical serologic herd classification for this study was based on the BioX 302, which primarily detects clinically ill animals [8,16]. The specificity under field conditions must therefore be investigated further, preferably in field studies based on best practice diagnostic test evaluation [17].

The BioX 302 showed a rather different test pattern. In general, most of the serum and milk samples tested negative. However, there were a small number of positive serum and milk samples in some herds (Figure 3 and Figure 4). The BioX 302 has been shown to have a poor sensitivity [5] and to primarily detect clinically ill animals [8,16]. Taking these findings into account, it is not surprising that the cows in this study generally tested negative when using the BioX 302, as only two herds were classified as having an ongoing *M. bovis* infection. The few positive test results were found in all three herd classifications (not infected, infected within the last 5 years and infected at sampling), and in particular, few positive results were found in milk samples from the non-infected herds (Figure 3 and Figure 4). Herd 4 was classified as not infected but had one positive serum sample and two positive milk samples, one of which was very high in ODC% (140). This could suggest subclinical mastitis in these cows, although they were not positive in PCR on milk. Herds 11 and 12 were classified as having an ongoing *M. bovis* infection, and both of these herds tested positive in a low number of serum and milk samples tested using BioX 302, and this was most pronounced in milk samples. Again, this could suggest subclinical mastitis cases in these herds. Based on the PCR samples from calves (Table 1), it seems that at least this age group was infected with *M. bovis* in herds 11 and 12. It would have been interesting to see the results of the BioX 302 on serum samples from calves—and whether this method would have detected the infection among calves. However, we have previously shown that the BioX 302 did not detect antibodies in calves exposed to *M. bovis* before 3 months of age [9], and all of the samples from the RC-project (Robust Calves project) are from calves under 3 months of age. Previous results have shown that the disease status among calves is not reflected in the bulk tank milk [18], and the findings of this study support that *M. bovis* infection in young stock cannot be measured using the BioX 302 in serum or milk samples from the cows either.

In comparison, there are very large differences in the test patterns between ID screen and BioX 302. Nearly all cows in all, but three non-infected herds were found to be positive when using the ID screen, while in contrast, very few cows tested positive using the BioX 302. As discussed above, the ID screen is a much more sensitive test than the BioX 302 and may be able to detect subclinically infected animals [3], as opposed to the BioX 302, which primarily detects diseased animals [8]. However, nearly all cows tested positive in herds without an ongoing infection, as well as in herds with no history of *M. bovis* infection. This leads to the hypothesis that the ID screen will measure exposure to *M. bovis* rather than colonization and dissemination of the organism in the infected animal. Whatever the reason for the very different test patterns, the interpretation and recommendations for the use of ID screen must be different from that of the BioX 302.

### 3.2. Correlation between Serum and Milk Samples

The concordance correlation coefficient between serum and milk S/P% was 0.66 (95% CI: 0.59–0.72). In general, the serum values were higher than the milk values, except in herd 12 (Figure 5). This could be explained by different clinical manifestations, e.g., clinical or subclinical mastitis cases could induce mostly high milk S/P% and systemic disease could induce mostly high serum S/P%, as previously shown when using the BioX 302 test [8]. This may also be the case for the ID screen, but to a lesser degree since the cows still test positive in both serum and milk.

The clinical signs present in the Outbreak-herds (herds 8–10) during the *M. bovis* disease outbreak are known as they were part of another *M. bovis* project. Herd 8 had many cows with clinical signs of arthritis and no *M. bovis* PCR-positive milk samples, while herds 9 and 10 experienced a combination of arthritis and mastitis among the cows. Herd 8 had clearly higher S/P% in serum than milk, while herds 9 and 10 had some very high milk values (Figure 5). Even though 4–5 years had passed since the *M. bovis* disease outbreak, the initial clinical expression may still be evident in the ID screen results. If this is the case, it is likely that herd 12 had subclinical mastitis cases.

The observed correlation between ID screen serum and milk values suggests that milk samples may be a promising replacement for serum samples. Strong responses observed in individual cows are also promising signs for the potential use of ID screen on bulk tank milk samples for herd-level diagnosis. This would be advantageous for surveillance and control purposes and for sampling many cows, since milk samples are easier and cheaper to collect than blood samples.

### 3.3. Uncertainty in Herd Classification

Herd classification is, among other things, based on previous BioX 302 tests and PCR on individual and bulk tank milk samples. The bulk tank milk samples were primarily collected as yearly surveillance tests and are therefore not sampled often enough to ensure the detection of new and mild infections– in the case of an *M. bovis* infection, the detectable response in bulk tank milk can be very short-lived for both BioX 302 and PCR [4,16]. It is possible that some of the herds could have had a previous *M. bovis* infection that was not detected in bulk tank milk by either BioX 302 or PCR, especially if the clinical signs were not severe and the farmer had not collected additional samples.

Herd 2 was classified as not infected despite one positive PCR sample and two positive BioX 302 serum samples, all from calves. The positive PCR test was one out of 209 tested samples. Taking into account that the PCR test is not 100% specific [19], this was judged to be a false positive result. Overall, based on the uncertainties in the diagnostic tests and no other indications of previous or current *M. bovis* infection, we have chosen to classify this herd as not infected.

The farmer from Herd 5 stated that the herd had experienced an *M. bovis* disease outbreak in 2012. It is possible that the ID screen would still be able to detect this exposure, even though seven years had passed since the disease outbreak. However, it is noteworthy that the calves did not test positive in the ID screen, despite calves often being the reservoir of the infection [3]. As discussed above, the BioX 302 is not a sensitive test in young calves, but an in-house ELISA with another antigen (MilA) has been evaluated with good sensitivity in young calves [9], indicating that another ELISA could perform better than the BioX 302 in calves. Based on the ID screen and PCR tests of calves, it seems likely that the young calves were not infected with *M. bovis*, and transmission must therefore occur among older calves in Herd 5. This implies that the milk management and separation of cows from young calves must be adequate in hindering transmission to the young calves. In this herd, the calves were born in a common calving pen and left with the cow for at least 12 h. The calves were then moved to single pens outside, very well separated from the cows. This management may have been sufficient to stop the young calves being exposed, even though there was infection among the cows. In Herd 4, none of the available tests were positive and the farmer stated that the herd had not had an *M. bovis* disease outbreak. This also highlights the difficulties in assessing infection with *M. bovis*. If the positive results of the ID screen are truly a sign of ongoing *M. bovis* infection or exposure within that herd, then it is a very difficult organism to detect. With nearly all cows testing positive in both serum and milk, we find it unlikely that these would be false positive samples. It could be that the ID screen cross-reacts with antibodies against other mycoplasma species. The importance of testing for cross-reactivity with other *Mycoplasma* spp., especially *M. agalactiae*, has been emphasized for other *M. bovis* ELISAs [20], however no such information can be found in the documentation for the ID screen [12,21].

## 4. Materials and Methods

### 4.1. Study Herds

Herds were selected for participation based on the availability of prior information about the *M. bovis* status. Information from dairy herds participating in two other research projects as well as knowledge from test results from previously collected samples made it possible to include herds known to have had an *M. bovis* outbreak and herds that had not had an outbreak (Table 1). Nine herds were included due to their participation in a large Danish calf-health research project (‘Robust calves project’ running from 2018–2021 in which nasal swabs, tracheal washes and blood samples were collected from randomly selected calves across three age groups). These herds are referred to as the RC-herds. The remaining three herds were included because they were known to have had an *M. bovis* disease outbreak with test-positive samples while participating in an *M. bovis* research project 3–4 years prior to the initiation of the present study [8]. These three herds are referred to as the Outbreak-herds.

There was variation among herds in how difficult it was to determine the present and previous *M. bovis* status and additional samples and diagnostic test history were therefore also included from on-farm animal health monitoring activities in order to facilitate the grouping of herds. Details are shown in Table 1.

Previous individual and bulk tank milk *M. bovis* PCR and ELISA results were confirmed from the Danish Cattle Database, which is a national cattle register for all Danish cattle herds. Both national surveillance and diagnostic tests voluntarily conducted on the request of the local veterinarian and farmers are registered here, and we included the available data from 2012–2019. PCR-tested milk samples from which results were available in the Danish Cattle Database were analyzed using the Pathoproof Major-3 or Complete-16 assays (Thermo Scientific, Waltham, MA) or Mastit 4 (DNA Diagnostic, Risskov, Denmark); the ELISA test used was the BioX 302.

Blood samples were collected from many calves in the RC-herds on several occasions during autumn and winter 2019 and 2020 as part of another project. To better characterize potential *M. bovis* infection in these herds, approximately 30 blood samples from seven of these herds were analyzed with the ID screen and BioX 302. During the RC-project, nasal swabs and tracheal washes were also collected from between 129 and 431 calves in each RC-herd (see Table 1 for details), and they were all tested for the presence of *M. bovis* with the Fluidigm PCR test (see Laboratory Analysis for details). No additional diagnostic tests were performed in the three Outbreak-herds.

On the basis of all the information gathered—and considering the fact that the sensitivity and specificity of the BioX 302 ELISA and the Fluidigm PCR tests are not perfect [5,6,19]—all herds were classified as either:

Not infected—meaning that none (or very few, likely false positives) of the available test results were positive for *M. bovis* and the farmer stated that they had never had clinical signs of *M. bovis-*associated disease or that the clinical signs occurred more than 5 years prior to sampling;Infected within the last 5 years—meaning that there were multiple positive diagnostic test results in previously or recently collected samples and/or reporting of clinical signs of *M. bovis* within the last 5 years prior to sampling;Infected at sampling—meaning that diagnostic tests indicated an ongoing infection with *M. bovis* among one or more age groups at the time of sampling for the present study.

### 4.2. Sample Collection

Paired serum and milk samples were collected from cows from the 12 dairy herds during the first quarter of 2019. For all nine RC-herds, paired blood and milk samples were collected from 20 lactating dairy cows, and a nasal swab was collected from the same cows for the detection of *M. bovis.* We aimed to collect samples from primiparous cows, but it was not practically feasible in all herds (Herds 1, 2 and 3), so older cows were also included in the sample collection. The blood samples were collected from the coccygeal vein in plain serum tubes. Prior to milk sampling, the teats were cleaned with ethanol on a tissue, the first milk was discarded and a composite milk sample consisting of milk from all udder quarters was collected from each cow in a bronopol-coated tube to preserve the milk sample. The nasal swabs were taken with a long sterile cotton swab, rubbed gently against the mucosa in one naris until saturated and placed in phosphate-buffered saline until analysis.

All procedures involving animals in this study were conducted in accordance with guidelines from the Danish Ministry of Justice with respect to animal experimentation and care of animals under study (The Danish Ministry of Justice, 2014, LBK no. 474). The Danish Animal Experiments Inspectorate under the Danish Veterinary and Food Administration was consulted for guidance on required permissions and approved the project activities in writing without requiring further formal application or approval processes. Following sampling of the animals, all herd owners were interviewed about their perception and experience with *M. bovis-*associated disease at their farm (summarized in Table 1).

Paired blood and milk samples were collected from 30 cows from the Outbreak-herds. No further tests were done in these herds as they were all known to have had a confirmed outbreak of *M. bovis-*associated disease in 2015–2016.

The number of cows included in each herd was optimized according to the available budget.

The serum and milk samples were analyzed for antibodies against *M. bovis* using the ID screen and the BioX 302. Following antibody analysis, the milk samples were frozen and stored for approximately 6 months and then tested for the presence of *M. bovis* bacterial DNA using the commercial PCR Pathoproof Major-3 assay. The nasal swabs were analyzed for the presence of *M. bovis* using the Fluidigm PCR system.

### 4.3. Laboratory Analysis

For the ID screen, a S/P% ≥ 60 for the serum sample was considered positive and for the milk samples the overnight incubation protocol was used in order to optimize the sensitivity and the samples were considered positive if S/P% ≥ 30 [12]. For the BIO K 302, an ODC% ≥ 37 was considered positive [22]. The Pathoproof Major-3 assay (Thermo Scientific, Waltham, MA, USA) and the Mastit 4 PCR assay (DNA diagnostic, Risskov, Denmark) were considered positive if the cycle threshold (Ct) value < 37. All these diagnostic tests were performed at Eurofins Milk Testing Denmark, Vejen, Denmark, according to the manufacturer’s instructions.

Nasal swabs from cows and calves and tracheal washes from calves were tested for the presence of *M. bovis* using the Fluidigm PCR system at the Technical University of Denmark, Lyngby, Denmark, and a Ct value < 30 was considered positive [19]. Cutoff values and test performance have not yet been established for this test.

### 4.4. Statistical Analysis

The correlation between serum and milk S/P% was calculated as the concordance correlation coefficient [23]. All data management and statistical analyses were carried out in R version 3.2.2 [24] using the packages “dplyr”, “ggplot2”, “gridExtra” and “DescTools”.

## 5. Conclusions

In the present study, we gained and shared experience of using the ID screen for the detection of antibodies against *M. bovis* in adult dairy cows under field conditions and compared this with the results of the BioX 302 test. When using the ID screen, nearly all cows in all, but three non-infected herds tested positive, while in contrast, very few cows tested positive when using the BioX 302. The ID screen is therefore a much more sensitive test than the BioX 302. However, some herds without ongoing infection, and even some herds with no history of *M. bovis* infection also tested positive. This indicates either lack of specificity (e.g., cross-reactions with other mycoplasma species) or that the ID screen measures exposure to *M. bovis* rather than the colonization and dissemination of the organism in the infected animal. The latter implies that the interpretation and recommendations for using the ID screen should be different from that of BioX 302. A concordance correlation coefficient between the ID screen serum and milk results of 0.66 (95% CI: 0.58–0.72) indicates that easy-to-collect milk samples can replace serum samples for ID screen diagnosis of *M. bovis* in adult cows, and the use of ID screen on bulk tank milk samples for surveillance and control purposes is promising. This, in addition to assessments of the ID screen performance (in particular regarding the specificity) under field conditions can provide new research questions to pursue. We therefore recommend field studies for best practice diagnostic test evaluation of the ID screen.

## Figures and Tables

**Figure 1 pathogens-09-00637-f001:**
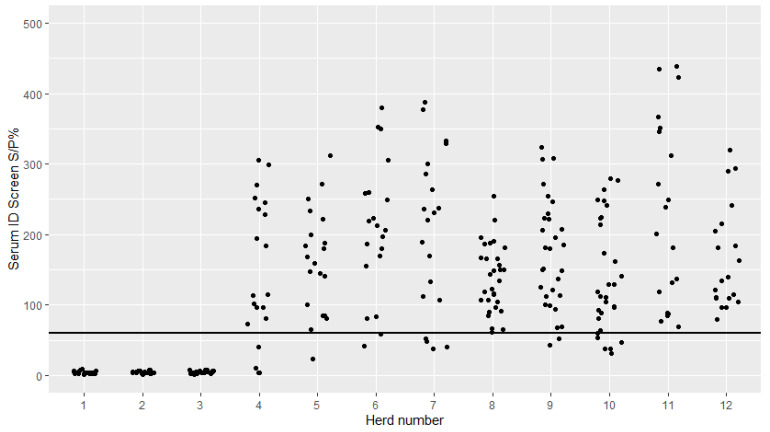
Serum antibody levels against *Mycoplasma bovis* in cows from 12 Danish dairy herds tested with the ID screen^®^ mycoplasma bovis indirect ELISA kit from IDvet. Herds 1–6 and 11–12 had 20 cows tested and herds 8–10 had 30 cows tested. The horizontal black line indicates the manufacturer-recommended cutoff value (sample-to-positive percentage (S/P%) of 60) for serum.

**Figure 2 pathogens-09-00637-f002:**
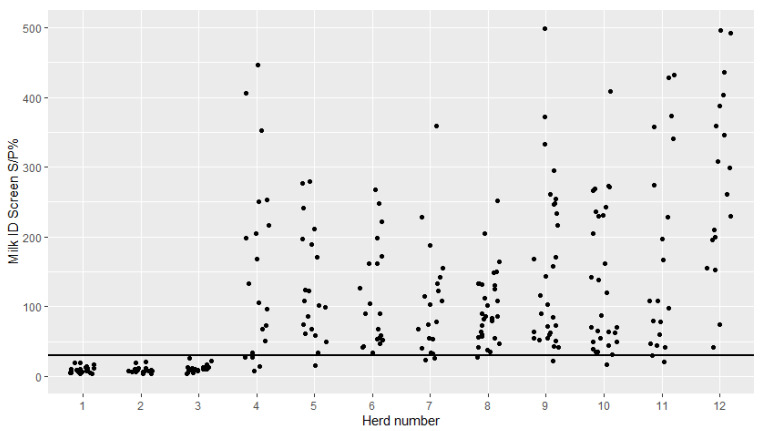
Milk antibody levels against *Mycoplasma bovis* in cows from 12 Danish dairy herds tested with the ID screen^®^ mycoplasma bovis indirect ELISA kit from IDvet. Herds 1–6 and 11–12 had 20 cows tested and Herds 8–10 had 30 cows tested. The horizontal black line indicates the manufacturer-recommended cutoff value (sample-to-positive percentage (S/P%) of 30) for the overnight protocol for milk.

**Figure 3 pathogens-09-00637-f003:**
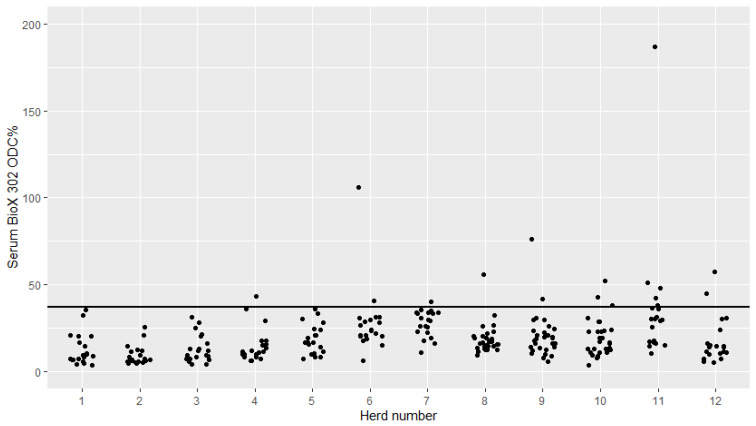
Serum antibody levels against *Mycoplasma bovis* in cows from 12 Danish dairy herds tested with the Bio K 302 ELISA kit from BioX. Herds 1–6 and 11–12 had 20 cows tested and Herds 8–10 had 30 cows tested. The horizontal black line indicates the manufacturer-recommended cutoff value (sample-coefficient (ODC%) of 37).

**Figure 4 pathogens-09-00637-f004:**
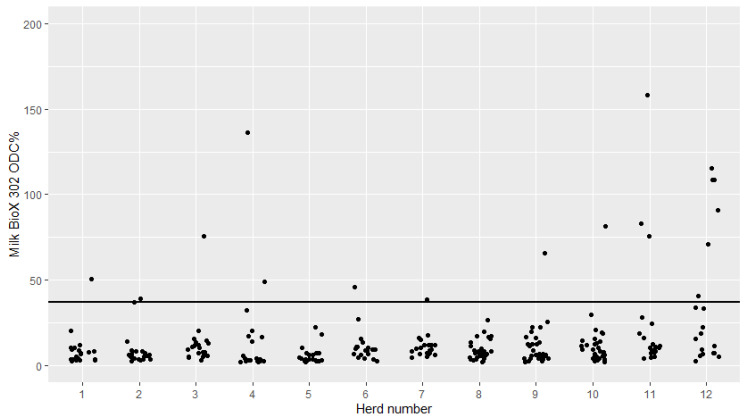
Milk antibody levels against *Mycoplasma bovis* in cows from 12 Danish dairy herds tested with the Bio K 302 ELISA kit from BioX. Herds 1–6 and 11–12 had 20 cows tested and Herds 8–10 had 30 cows tested. The horizontal black line indicates the manufacturer-recommended cutoff value (sample-coefficient (ODC%) of 37).

**Figure 5 pathogens-09-00637-f005:**
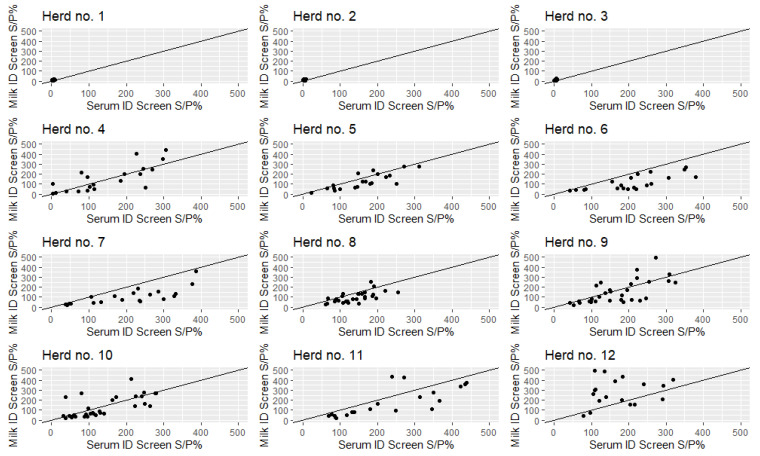
Correlations between paired serum and milk sample-to-positive percentage (S/P%) against *Mycoplasma bovis* using the ID screen^®^ mycoplasma bovis indirect ELISA kit, stratified across 12 Danish dairy herds. The diagonal line indicates perfect agreement between serum and milk values.

**Table 1 pathogens-09-00637-t001:** Overview of the herd type and size, previous *Mycoplasma bovis-*associated disease history, as well as previous diagnostic test results (from 2012–2019) and results of additional samples taken (RC-Calves in 2019–2020) for *Mycoplasma bovis* classification of 12 Danish dairy cattle herds (RC = Robust calves project, N/A = not available, BTM = bulk tank milk).

Herd No.	Herd Type	No. of Cows ^a^	PCR—Individual Cows ^b^ (Positives/n)	ELISA—Individual Cows/ Calves ^c^ (Positives/n)	BTM PCR ^b^ (Positives/n)	BTM ELISA ^c^ (Positives/n)	RC-Calves—ID Screen ^d^ (Positives/n)	RC-Calves—BioX 302 ^c^ (Positives/n)	RC-Calves—PCR Test ^e^ (Positives/n)	*Mycoplasma bovis* Disease Outbreak	*Mycoplasma bovis* Classification
**1**	RC	150	N/A	N/A	0/21	0/16	0/27	1/27	0/182	No ^f^	Not infected
**2**	RC	190	0/348	N/A	0/14	0/8	0/27	2/27	1/209	No ^f^	Not infected
**3**	RC	350	0/9	0/8	0/9	1/5	0/29	1/29	0/228	Yes (2013) ^f^	Not infected
**4**	RC	220	0/1	N/A	0/10	0/6	0/29	0/29	0/172	No ^f^	Not infected
**5**	RC	200	0/3	N/A	0/11	1/6	1/29	0/30	0/129	Yes (2012) ^f^	Not infected
**6**	RC	700	1/398	N/A	0/9	2/5	16/30	2/30	3/431	No ^f^	Infected within the last 5 years
**7**	RC	600	0/284	N/A	1/23	0/6	24/30	10/30	1/179	Yes (2014–2015) ^f^	Infected within the last 5 years
**8**	Outbreak	190	7/140	85/372	0/34	0/23	N/A	N/A	N/A	Yes (2015–2016) ^g^	Infected within the last 5 years
**9**	Outbreak	430	69/1188	70/282	18/327	1/9	N/A	N/A	N/A	Yes (2015–2016) ^g^	Infected within the last 5 years
**10**	Outbreak	200	21/98	91/303	3/16	1/12	N/A	N/A	N/A	Yes (2015–2016) ^g^	Infected within the last 5 years
**11**	RC	600	10/25	0/3	0/10	1/7	N/A	N/A	11/256	Yes (2014) ^f^	Infected at sampling
**12**	RC	330	4/234	N/A	0/14	0/6	N/A	N/A	9/228	No ^f^	Infected at sampling

^a^ Average number of cows per year during 2016–2019 ^b^ Pathoproof Major-3 or Complete-16 assays (Thermo Scientific, Waltham, MA, USA) or Mastit 4 (DNA diagnostic, Risskov, Denmark), test results collected on the farmers’ initiatives prior to this study (e.g., for herd health management), positive cycle threshold < 37 ^c^ BioX 302 (BioX Diagnostics, Rochefort, Belgium), test results collected on the farmers’ initiatives prior to this study (e.g., for herd health management), positive ≥ 37 sample coefficient (ODC%) ^d^ ID screen^®^ mycoplasma bovis indirect (IDvet, Grabels, France), positive ≥ 60 sample-to-positive percentage (S/P%) ^e^ Fluidigm In-house PCR (Technical University of Denmark), positive cycle threshold < 30. ^f^ According to the farmer ^g^ Confirmed test-positive for *M. bovis*.

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
