# Peer review of "Field Experience of Antibody Testing against Mycoplasma bovis in Adult Cows in Commercial Danish Dairy Cattle Herds"

_pathogens, 2020, doi:10.3390/pathogens9080637_

Round 1

Reviewer 1 Report

This study has evaluated the performance of two commercially available antibody ELISAs for Mycoplasma bovis. Overall, the study is interesting, however many of the results provided are qualitative in nature which makes it difficult to draw conclusions from the study datasets. This evident throughout the text with many generalised statements. As an example, on line 110, the authors state: “The overall correlation between serum- and milk S/P% was relatively good at 0.67.” One might ask, relative to what?

A key question is why the datasets were not analysed in a more robust statistical framework? Perhaps, this was due to the Bio K 302 giving so few positive results, if so, this should be clearly stated.

With respect to the ID Screen, did the authors consider calculating sensitivity and specificity estimates based on the known (or imputed) exposure status of the herds?

Overall, the manuscript suggests the authors are not convinced by their dataset with many examples of qualified terms and generalised conclusions. In revising the manuscript, the authors should develop a very clear objectives for their study and ensure these are carried (and answered) through the document. Where the objective might be answerable by the datasets, then some more speculative elements could be added on how this might be achieved.

Specific comments and suggestions:

Line 20: Please reconsider the sentence: “The ID Screen seems to be a much more sensitive test than the BioX 302.”

The sentence is very ambiguous, the authors have elected to conduct the study and write the paper, thus are in many ways obligated to make conclusions based on their datasets.

Line 29 suggest replacing “causing” with “associated with”

Line 54 suggest replacing “homologous” with “similar”. The term “homologous” in this context refers to two genes which share a common ancestral gene. Therefore, they either do or do not, so the answer is either yes or no respectively.

Line 76 suggest revision “the Danish dairy industry” – the current sentence seems incomplete.

Line 110: Was the “overall correlation” determined from a combined dataset?

Line 111 suggest revision “within each herd is show in Figure 5.”

Line 112 suggest revision “correlations are not shown”

Line 121 Are the calculated correlations provided in the manuscript? I could not locate them.

Line 142 The authors should be cautious using terms such as “well above the cut-off”, technically, any sample above the cut-off is positive, we used these values to prevent subjective assessment of data.

Line 154 Does the manufacturer of the ID Screen claim being able to differentiate healthy and diseased cows? Most serological based tests are really aiming to test for exposure unless they include the capacity to detected and differentiate between IgG and IgM.

Line 156 these types of qualitative statements are problematic. Assessing a test as being “very sensitive” based on the number of samples being “well beyond the cut-off” is not the correct way this should be determined. As mentioned previously, any sample which passes the threshold is positive. Sensitivity, and for that matter specificity, should be calculated based on the results of testing a panel of samples of known exposure status to the pathogen of interest.

Line 200 Could the authors please clarify what they mean by “predilection sites of the cows.”

Line 212 again “The relatively good correlation between” should be more specific.

Line 266-268 Please review the following statement, its meaning is unclear:

“Because it was different how easy it was to evaluate the present and current M. bovis status in the herds, different additional available test results were included. Details are shown in Table 1.”

Author Response

Reviewer 1:

Comments and Suggestions for Authors

This study has evaluated the performance of two commercially available antibody ELISAs for Mycoplasma bovis. Overall, the study is interesting, however many of the results provided are qualitative in nature which makes it difficult to draw conclusions from the study datasets.

AU: thank you for pointing this out. We see that the term ‘performance’ can be misunderstood and raise expectations of Se and Sp-calculations being performed and presented in the study. The dataset available for this study is not suitable for diagnostic Se-Sp calculations according to best practice guidelines on diagnostic test evaluation (OIE: https://www.oie.int/doc/ged/D12069.PDF), mainly due to a low number of test-positives in BioX 302 and a low number test-negatives in ID Screen, but also due to the lack of diagnostic test-results from test with other biological mechanisms (e.g. bacteriological culture) at cow-level. We have therefore rephrased the objectives to reduce the risk of misunderstandings. Lines 82-86: “The objectives of this study were therefore to: 1) gain and share experience of using the new commercial ELISA ID Screen for the detection of antibodies against M. bovis in adult dairy cows under field conditions, 2) determine the correlation between the measured antibody levels in milk and serum, and 3) compare the ID Screen results with the results of the frequently used and commercially available BioX 302 ELISA.” This has been corrected throughout the paper.

This evident throughout the text with many generalised statements. As an example, on line 110, the authors state: “The overall correlation between serum- and milk S/P% was relatively good at 0.67.” One might ask, relative to what?

AU: We have recalculated the correlation and rephrased the reporting and interpretation of the results.

A key question is why the datasets were not analysed in a more robust statistical framework? Perhaps, this was due to the Bio K 302 giving so few positive results, if so, this should be clearly stated.

AU: We understand this request, and of course the data COULD BE analysed using for instance a generalized mixed regression model to compare the probability of testing positive between the test method and herd status categories. However, the results are crystal clear; there is a huge difference between the test results between ID Screen and BioX 302 as well as between the test-results from cows from non-infected vs. the infected/prior infected herds. We do not think that carrying out a statistical analysis will add more value to the study or the interpretation of the results.

With respect to the ID Screen, did the authors consider calculating sensitivity and specificity estimates based on the known (or imputed) exposure status of the herds?

AU: We did consider it, but chose not to, for the reasons listed above. The data are not suitable for such calculations. However, based on what we have observed, we recommend that a study is carried out under field conditions to determine the Se and, in particular, the Sp of the ID Screen ELISAs at animal or group level. This is now specified in the manuscript e.g. lines 190-91: “The specificity under field conditions must therefore be investigated further, preferably in field studies based on best practice diagnostic test evaluation.”

Overall, the manuscript suggests the authors are not convinced by their dataset with many examples of qualified terms and generalised conclusions. In revising the manuscript, the authors should develop a very clear objectives for their study and ensure these are carried (and answered) through the document. Where the objective might be answerable by the datasets, then some more speculative elements could be added on how this might be achieved.

AU: we have included a more clear description of the objectives and made sure to answer the objectives directly and more clearly in the manuscript.

Specific comments and suggestions:

Line 20: Please reconsider the sentence: “The ID Screen seems to be a much more sensitive test than the BioX 302.”

The sentence is very ambiguous, the authors have elected to conduct the study and write the paper, thus are in many ways obligated to make conclusions based on their datasets.

AU: That is correct, thanks for the comment. In line 21 we have changed to “….is therefore a much more…”.

Line 29 suggest replacing “causing” with “associated with”

AU: has been replaced.

Line 54 suggest replacing “homologous” with “similar”. The term “homologous” in this context refers to two genes which share a common ancestral gene. Therefore, they either do or do not, so the answer is either yes or no respectively.

AU: has been replaced.

Line 76 suggest revision “the Danish dairy industry” – the current sentence seems incomplete.

AU: The sentence has been changed to “The potential use of ELISA on milk samples to classify or monitor dairy herds for M. bovis infection will be of interest in a setting like the Danish dairy industry, since the sampling can be automated via the mandatory milk quality control scheme and bulk tank milk surveillance for other cattle diseases.”, lines 78-81.

Line 110: Was the “overall correlation” determined from a combined dataset?

AU: Yes, we calculated the correlation based on a dataset in which laboratory results from the serum and milk samples collected from the same animals on the same day (paired) were included. Lines 125-126 has been changed to: “The Concordance Correlation Coefficient between serum and milk S/P% across the full dataset of paired serum and milk samples was 0.66 (95%CI: 0.59-0.72).”

Line 111 suggest revision “within each herd is show in Figure 5.”

AU: has been changed in line 127.

Line 112 suggest revision “correlations are not shown”

AU: has been changed in line 128.

Line 121 Are the calculated correlations provided in the manuscript? I could not locate them.

AU: No, the correlation for each herd is not provided in the manuscript. We find that there were too few cows in each herd for a correlation calculation in each herd, but the graphs add some value in the discussion of different patterns across herds.

Line 142 The authors should be cautious using terms such as “well above the cut-off”, technically, any sample above the cut-off is positive, we used these values to prevent subjective assessment of data.

AU: thanks for the comment, we agree, it is more correct not to add this subjective assessment. We have deleted the term “well above the cut-off” several places in the manuscript.

Line 154 Does the manufacturer of the ID Screen claim being able to differentiate healthy and diseased cows? Most serological based tests are really aiming to test for exposure unless they include the capacity to detected and differentiate between IgG and IgM.

AU: No, they do not, but this is seen in relation to the BioX 302 which is shown to test positive in diseased animals (Petersen et al 2018). This is elaborated in the section from lines 173-193.

Line 156 these types of qualitative statements are problematic. Assessing a test as being “very sensitive” based on the number of samples being “well beyond the cut-off” is not the correct way this should be determined. As mentioned previously, any sample which passes the threshold is positive. Sensitivity, and for that matter specificity, should be calculated based on the results of testing a panel of samples of known exposure status to the pathogen of interest.

AU: The sentence rephrased to “The ID Screen is a sensitive test, as all or nearly all cows in each herd tested either positive or negative.” In lines 182-83. See in addition our response to your general comments above.

Line 200 Could the authors please clarify what they mean by “predilection sites of the cows.”

AU: Predilections sites have been deleted and the sentence rephrased to “This could be explained by different clinical manifestations, e.g. clinical or subclinical mastitis cases could induce mostly high milk S/P%, and systemic disease could induce mostly high serum S/P%, as previously shown when using the BioX 302 test.” In lines 230-235.

Line 212 again “The relatively good correlation between” should be more specific.

AU: The sentence has been rephrased in lines 244-45: “The observed correlation between ID Screen serum and milk values suggests that milk samples may be a promising replacement for serum samples.”

Line 266-268 Please review the following statement, its meaning is unclear:

“Because it was different how easy it was to evaluate the present and current M. bovis status in the herds, different additional available test results were included. Details are shown in Table 1.”

AU: We have rephrased the sentence in lines 299-306: “There was variation among herds in how difficult it was to determine the present and previous M. bovis status, and additional samples and diagnostic test history were therefore also included from on-farm animal health monitoring activities in order to facilitate the grouping of herds. Details are shown in Table 1.”

As suggested, we have had the manuscript proofread by a native English speaker and believe it has improved a lot.

Reviewer 2 Report

Summary:

The manuscript describes a small field study in the diagnostics of Mycoplasma bovis. The aim of this study was to evaluate the performance of the ID Screen® Mycoplasma bovis Indirect (ID Screen) from IDvet in France for detection of antibodies against M. bovis in adult dairy cows in serum and blood samples. An additional aim was to compare the results with the performance of Bio K 302 (BioX 302) from BioX Diagnostics in Belgium. Another additional objective was to assess the correlation between the measured antibody levels in milk and serum. The ID Screen tested nearly all cows positive in all herds with exception of three non-infected herds. In contrast, the BioX 302 tested very few cows positive. The authors conclude that the ID Screen seems to be a much more sensitive test than the BioX 302. However, some herds without on-going infection, and even herds without a history of M. bovis infection also tested positive in the ID Screen. Therefore, the interpretation and recommendations for the use of ID Screen must be different from that of BioX 302. The correlation between serum- and milk values for the ID Screen good enough to be promising for the use of milk samples instead of serum samples

General remarks:

The manuscript is very well elaborated and shows already a high scientific level. From the introduction to the presentation of results, M&M section to and the discussion my impression is favourable. Therefore, only very few points are left to raise for reviewers. My compliments to the authors.

Detailed remarks:

Introduction

L41-42: Has only the sensitivity of BioX 302 been reported to be low or also the specificity? Please clarify.

L47: I think the comma following “this” is not necessary.

Results

L84-86: Due to the order of the manuscript with M&M at the end, table 1 should be referenced here in the text. Otherwise, RC-herds are a mystery for the reader.

Discussion

L207 and L210: Replace “have” by “has”.

L247: Replace “is” by “are” before “false positive samples”.

M&M

I found no reasoning for the study design as well as the numbers of herds and cows investigated. Please add this information.

Table 1: There are a lot of PCR and BTM PCR tests stated for herd 9. Are the figures correct? For herds 11 and 12, no serological results are available for RC-Calves. I wonder why, but maybe it was somewhere mentioned in the text and I simply did not spot it.

L335: Please state the R packages used (if so).

Author Response

Reviewer 2:

General remarks:

The manuscript is very well elaborated and shows already a high scientific level. From the introduction to the presentation of results, M&M section to and the discussion my impression is favourable. Therefore, only very few points are left to raise for reviewers. My compliments to the authors.

AU: Thanks for positive comments. Answers to specific comments are described below.

Detailed remarks:

Introduction

L41-42: Has only the sensitivity of BioX 302 been reported to be low or also the specificity? Please clarify.

AU: We meant that the BioX only had a low sensitivity, but we have now added the specificities in lines 44-47: “The Bio K 302 ELISA kit (BioX 302) has been reported to have low sensitivity ranging from 0.37-0.50 and specificity ranging from 0.90-0.96 in experimental studies [5–7], and very short-lasting antibody detection in individual cows [8] and calves [9].”

L47: I think the comma following “this” is not necessary.

AU: the comma has now been deleted. 

Results

L84-86: Due to the order of the manuscript with M&M at the end, table 1 should be referenced here in the text. Otherwise, RC-herds are a mystery for the reader.

AU: A sentence has been added to clarify this in lines 94-95. “See Table 1 and the Materials and Methods section for a description of the different herds included in the study.”

Discussion

L207 and L210: Replace “have” by “has”.

AU: has been replaced.

L247: Replace “is” by “are” before “false positive samples”.

AU: Has been replaced.

M&M

I found no reasoning for the study design as well as the numbers of herds and cows investigated. Please add this information.

AU: To have knowledge about the M. bovis disease status in the herds, it was convenient that we knew the herds from other projects, which was the case for the 12 herds included in the study. So the number of herds are based on access to herds with possibilities of investigating their M bovis status. The number of cows sampled in each herd, was optimized according to the available budget. This is elaborated in lines 288-92: “Herds were selected for participation based on the availability of prior information about the M. bovis status. Information from dairy herds participating in two other research projects as well as knowledge from test results from previously collected samples made it possible to include herds known to have had an M. bovis outbreak and herds that had not had an outbreak (Table 1). and line 367: “The number of cows included in each herd, was optimised according to the available budget.”

Table 1: There are a lot of PCR and BTM PCR tests stated for herd 9. Are the figures correct? For herds 11 and 12, no serological results are available for RC-Calves. I wonder why, but maybe it was somewhere mentioned in the text and I simply did not spot it.

AU: Yes, the figures are correct. The PCR and BTM PCR samples are voluntary samples taken by the farmer for e.g. surveillance and dry-off. Some herds, such as herd 9 and probably also 2, 6 and 12, use these samples to dry-off cows while other herds use maybe bacterial culture, that is not reported to the Danish Cattle Database and therefore we did not have access to these samples for this study. There were no additional antibody tests of calves in the Outbreak-herds (8, 9 and 10), which is now described in lines 319-20: “No additional diagnostic tests were performed in the three Outbreak-herds.”

L335: Please state the R packages used (if so).

AU: this has now been elaborated in lines 287-88:“, using the packages “dplyr”, “ggplot2”, “gridExtra”, and “DescTools”.”

Reviewer 3 Report

Paper really innovative, calibrated and well - built text.

I didn't immediatly understand the results at the beginning of the work , but I really liked the general approach , the insertion of the control cards and the diagnostic sets and wel indicated and rendered. 

I believe this field experience important and useful for routine work.

Author Response

Reviewer 3:

AU: Thanks for the positive comments. We have had the manuscript proofread by a native English speaker and believe it has improved a lot.

Round 2

Reviewer 1 Report

The authors have adequately addressed the concerns and comments raised in my initial review of their manuscritp.